# Influence of Cushion Plant *Androsace tapete* on Nitrogen Uptake Strategies of Associated Alpine Plants

**DOI:** 10.3390/plants14203232

**Published:** 2025-10-21

**Authors:** Shuo Xing, Yong-Tao He, Pei-Li Shi, Xing-Liang Xu

**Affiliations:** 1Lhasa Plateau Ecosystem Research Station, Key Laboratory of Ecosystem Network Observation and Modeling, Institute of Geographic Sciences and Natural Resources Research, Chinese Academy of Sciences, Beijing 100101, China; xingshuo0650@igsnrr.ac.cn (S.X.); shipl@igsnrr.ac.cn (P.-L.S.); 2University of Chinese Academy of Sciences, Beijing 100049, China; 3College of Resources and Environment, University of Chinese Academy of Sciences, Beijing 100190, China; 4Key Laboratory of Ecosystem Network Observation and Modeling, Institute of Geographic Sciences and Natural Resources Research, Chinese Academy of Sciences, Beijing 100101, China

**Keywords:** *Androsace tapete*, facilitation, ^15^N labeling, N uptake pattern, Qinghai–Tibet Plateau

## Abstract

In alpine ecosystems, plant growth is often constrained by multiple environmental factors, especially the infertile soils with lower temperature that decelerate the rate of nutrient turnover, thus leading to a diminished availability of nutrients in the soil, notably nitrogen (N), and its different forms, which is a pivotal factor for limiting plant growth and species coexistence in these alpine areas. *Androsace tapete* (*A. tapete*) is an endemic species and the most widely distributed cushion plant on the Qinghai–Tibet Plateau (QTP). Its positive interactions can facilitate other associated plants to deal with severe environmental conditions in the alpine grassland ecosystem. The change in soil nutrient availability is one of the main positive interactions, but little is known about how *A. tapete* changes soil nutrient availability and affects the N uptake pattern of associated plants. This study investigated the N utilization patterns of three associated plant species —*Carex atrofusca* (*C. atrofusca*), *Cyananthus incanus *(*C. incanus*), and *Potentilla saundersiana *(*P. saundersiana*)— growing inside the cushion area *A. tapete* (CA) and the ambient grassland without cushion plants (CK), using a ^15^N labeling method to clarify the effect of *A. tapete* on the N uptake strategies with NH_4_^+^, NO_3_^−^, and organic N of its associated species. The results showed the following: (1) compared to CK, the soil total C, total N, and available NH_4_^+^ contents under the *A. tapete* showed a significant 47.82%, 40.96%, and 47.33% increase, respectively; (2) *A. tapete* showed a stronger preference for NH_4_^+^ (>80%), whereas the associated species in CK exhibited a more balanced uptake, deriving 39.29–55.59% of N from NO_3_^−^, 25.72–44.00% from NH_4_^+^, and 16.15–18.69% from glycine. (3) The three associated plants possessing *A. tapete* significantly reduced their uptake of glycine by 9.76%, 12.55%, and 7.15%, respectively, while the absorption of NH_4_^+^ by *C. atrofusca* and *C. incanus* increased by 18.46% and 36.11%; meanwhile, NO_3_^−^ uptake decreased by 8.70% in *C. atrofusca* and 23.55% in *C. incanus*. These findings indicated that the *A. tapete* can change the N uptake pattern of the associated plants growing inside the cushion body, such as enhancing the absorption of inorganic N and decreasing the organic N. This adaptive strategy of the associated plants with cushion plant enables them to counteract the N-limited conditions prevalent in alpine environments, and, as a consequence, facilitates their growth and promotes local plant community diversity in the alpine environment.

## 1. Introduction

The Qinghai–Tibet Plateau (QTP) is one of the most unique and ecologically fragile alpine regions on Earth, characterized by extreme climatic conditions such as low temperatures, high radiation, and nutrient-poor soils [1,2]. Among these environmental constraints, nitrogen (N) availability is widely recognized as the principal limiting factor for plant growth in this region [3]. Due to limited weathering, slow organic matter accumulation, and persistently low temperatures, microbial activity and key biogeochemical processes such as nitrogen fixation and mineralization are strongly suppressed [4,5,6]. Consequently, the overall availability of soil nitrogen is substantially reduced, thus intensifying nutrient stress and shaping plant communities in alpine ecosystems.

To survive under such nutrient-deficient conditions, alpine plants have evolved flexible nitrogen acquisition strategies, enabling them to utilize multiple forms of nitrogen [7]. In addition to inorganic sources such as ammonium (NH_4_^+^) and nitrate (NO_3_^−^), many species can directly absorb low-molecular-weight organic compounds such as amino acids [8]. Studies have shown that plant preferences for nitrogen forms can shift depending on soil nitrogen availability [9]. For instance, Xu et al. found that dominant alpine species increased their uptake of nitrate while reducing reliance on organic nitrogen under localized nutrient enrichment [10]. This plasticity in nitrogen acquisition plays a vital role in alleviating nutrient competition, promoting niche differentiation, and enhancing species coexistence within plant communities [11,12].

Cushion plants, a prominent functional group in alpine ecosystems, are well known for their facilitative roles and capacity as ecosystem engineers. With their compact growth form, dense canopy, and long life span, cushion plants can significantly modify their surrounding microhabitat—buffering temperature fluctuations, retaining moisture, and, most importantly, altering soil nutrient dynamics [13,14,15]. Numerous studies have reported that cushion plants can enhance soil nitrogen availability by stimulating microbial and enzymatic activity, accelerating nitrogen mineralization, and increasing inorganic nitrogen concentrations [16,17,18,19]. Moreover, their ability to trap organic litter leads to elevated organic matter pools and nutrient levels beneath their canopies compared to adjacent open areas [20,21,22,23,24].

These positive interactions among cushion plants not only create favorable conditions for other plants' growth but also promote the coexistence of different species within their cushion body, ultimately contributing to higher biodiversity in alpine communities. Among the various mechanisms driving this facilitation, changes in soil nutrient availability are considered fundamental [17,25]. However, most existing studies have focused on the effect of cushion plants on nutrient supply or availability, while little research focuses on how these changes influence the nutrient uptake strategies of the associated plants within the cushion microhabitat [19,22,26]. As one of the most dominant cushion species on the QTP, *Androsace tapete* functions as a keystone ecosystem engineer, capable of profoundly altering soil microclimate and nutrient cycling processes that shape community assembly and persistence.

Building on this, we hypothesize that *A. tapete* modifies nitrogen uptake strategies in associated species by altering nitrogen availability, particularly enhancing inorganic nitrogen uptake (such as NH_4_^+^) and reducing reliance on organic nitrogen sources in the cushion microhabitat. To test this hypothesis, we investigated whether *A. tapete*—one of the most widespread and ecologically dominant cushion species on the QTP—modifies nitrogen uptake strategies in their associated alpine plants. Specifically, we used the ^15^N stable isotope tracing method to compare the uptake of three major nitrogen forms—NH_4_^+^, NO_3_^−^, and glycine—by three dominant associated species (*Carex atrofusca*, *Cyananthus incanus*, and *Potentilla saundersiana*) growing either inside the cushion area *A. tapete* (CA) or the ambient grassland without cushion plants (CK).

## 2. Results

### 2.1. Soil Analysis

Compared to the CK, the soil total C, total N, and NH_4_^+^ content in CA were significantly increased by 47.82%, 40.96%, and 47.33%, respectively (Table 1). The soil moisture content, C/N ratio, NO_3_^−^, and glycine content under the cushion plant were 5.62%, 4.92%, 7.89%, and 14.28% higher than that of CK, respectively, but did not show a significant difference. The soil bulk density in CA was significantly lower than that of CK, with a reduction of 40.95%, while the pH showed no significant difference.

### 2.2. Biomass Differences

As shown in Table 2, the leaf and root biomass of the three associated plant species differed between the CA and CK. Overall, the leaf biomass of all three species tended to be higher in CA than in CK, whereas root biomass generally showed the opposite trend. Specifically, the leaf biomass of *C. atrofusca* in CA was significantly higher than that in CK, with CA being approximately 2.39 times that of CK. In contrast, *C. incanus* and *P. saundersiana* also exhibited higher leaf biomass in CA, with an increase of 13.95% and 20.00%, respectively, but these differences were not statistically significant.

For root biomass, a significant difference was observed only in *C. atrofusca*, with CK having 2.21 times greater biomass than that of CA. No significant differences were found in the root biomass of *C. incanus* and *P. saundersiana*, although both species had higher root biomass in CK, with values approximately 1.45 and 1.92 times that of CA, respectively.

### 2.3. N Uptake Differences

In CA plots, *A. tapete* predominantly absorbed nitrogen in the form of NH_4_^+^ (82.84%), with much lower proportions of NO_3_^−^ and glycine, indicating a clear preference for ammonium. In contrast, the associated species displayed more diverse nitrogen uptake patterns. *C. atrofusca* absorbed comparable amounts of NO_3_^−^ (46.89%) and NH_4_^+^ (44.18%), with glycine accounting for a small fraction (8.93%). *C. incanus* relied heavily on NH_4_^+^ (72.31%) but also absorbed NO_3_^−^ (24.09%) and a small amount of glycine (3.60%). *P. saundersiana* exhibited an intermediate pattern, with NH_4_^+^ as the dominant nitrogen form (51.53%), followed by substantial NO_3_^−^ uptake (38.91%) and moderate glycine uptake (9.56%).

Comparisons between CA and CK revealed species-specific shifts in nitrogen uptake. For *C. atrofusca*, glycine uptake increased significantly, with a 9.76% rise, while NH_4_^+^ uptake decreased by 18.46%, and NO_3_^−^ uptake remained stable. For *C. incanus*, there was a marked increase in glycine uptake (up by 12.55%) and NO_3_^−^ uptake (up by 23.55%), while NH_4_^+^ uptake decreased by 36.11%. In *P. saundersiana*, only glycine uptake increased, with a 7.15% rise, while NO_3_^−^ and NH_4_^+^ remained relatively stable, with minimal changes in both (Table 3).

## 3. Discussion

### 3.1. Differences in Nitrogen Form Uptake

Plant species diversity and coexistence are central topics in community ecology. Previous studies suggest that coexistence is often facilitated by resource partitioning, where species differ in their ability to exploit particular resources [11,27,28]. In N-limited alpine ecosystems, mechanisms such as distinct nutrient requirements and variation in N uptake strategies across time, space, and chemical forms play important roles in supporting species coexistence [10,29,30]. In these harsh environments characterized by low temperature and slow mineralization rates, nitrogen is a key limiting nutrient for plant growth, and scarcity generally intensifies interspecific competition [22,31]. Cushion plants, however, act as ecosystem engineers that can improve local soil conditions. Our results indicate that *A. tapete* increased the availability of inorganic N, especially NH_4_^+^, which is the dominant and most stable form under alpine conditions, thereby altering not only its own uptake but also that of associated species and ultimately creating opportunities for niche differentiation [32,33,34].

Within cushion patches, species displayed clear contrasts in N uptake strategies. *A. tapete* relied overwhelmingly on NH_4_^+^ (82.84%), reflecting a strategy of maximizing use of the most abundant and readily available form. By comparison, associated species exhibited more diverse profiles. *C. atrofusca* showed the highest proportion of NO_3_^−^ uptake (46.89%) while simultaneously absorbing comparable amounts of NH_4_^+^ (44.18%), indicating a flexible strategy that reduces reliance on a single form. *C. incanus* absorbed large amounts of NH_4_^+^ (72.31%) but also maintained notable NO_3_^−^ uptake (24.09%), suggesting partial overlap with *A. tapete* alongside complementary nitrate use. *P. saundersiana* exhibited intermediate uptake of both inorganic forms (NO_3_^−^ 38.91%; NH_4_^+^ 51.53%), reflecting a balanced strategy between ammonium and nitrate acquisition. In addition, all associated species incorporated small proportions of organic N through glycine (3.60–9.56%).

These differences suggest that while *A. tapete*, as a dominant and highly productive cushion species, concentrates on the primary N form (NH_4_^+^), and associated species partition their uptake across NH_4_^+^, NO_3_^−^, and organic N, thereby alleviating direct competition and promoting complementary resource use [17,18,32,35]. Such differentiation aligns with findings from Arctic tundra systems, where species differ in N acquisition strategies depending on timing, rooting depth, and chemical form of available N [36,37]. In our system, the pattern that the dominant cushion species preferentially exploits NH_4_^+^ while co-occurring species distribute their uptake among multiple N forms reflects a functional trade-off between specialization and flexibility. This trade-off reduces competitive exclusion, enhances facilitative interactions, and contributes to the maintenance of species diversity and community stability in alpine environments [10,38].

The observed differences in nitrogen uptake among the species in our study may be further influenced by several factors, including bulk density. The decrease in bulk density under the cushions implies an increase in soil porosity and aeration, which are conditions that often enhance microbial activity and N mineralization [39]. These changes likely contribute to the observed increase in inorganic nitrogen [40]. The stress-gradient hypothesis suggests that facilitation by cushion plants is more pronounced under harsh conditions, where resources are limiting, but as resources improve, competition among species may increase [41]. This aligns with the shift in nitrogen uptake we observed in the associated species, which were able to partition their nitrogen sources more effectively in the cushion microhabitats.

With climate warming, faster mineralization could increase background inorganic nitrogen levels, potentially reducing the relative advantage of cushion plants by amplifying resource availability for all species [42]. However, the microclimatic buffering effect provided by cushions, such as moderating temperature extremes, would likely remain an important facilitative mechanism in these systems [17]. Moreover, microbial processes, including enzyme activity and net nitrogen mineralization rates, likely play a key role in regulating nutrient dynamics within cushion patches, further promoting the shift in nitrogen uptake patterns we observed [35]. This process of niche differentiation is comparable to patterns observed in other extreme environments, such as the Arctic tundra, where species utilize different timing, rooting depths, and nitrogen forms to coexist [43].

### 3.2. Reduced Dependence on Organic N

In CK, the proportion of glycine uptake by associated plants was much higher than that in CA plots, ranging from 16.15% to 18.69%, compared to only 3.60% to 9.56% in CA. Low temperatures suppress microbial activity and slow the decomposition of plant litter and organic matter, thereby inhibiting N mineralization and reducing the availability of inorganic N sources such as NO_3_^−^ and NH_4_^+^ [17]. In response, alpine plants in CK tend to rely more on organic N sources such as amino acids (e.g., glycine) to fulfill their nutritional requirements. This shift highlights a key adaptive strategy that enables plant communities to maintain function and persistence under nutrient-limited conditions [32,44].

In contrast, plants growing within the cushion body of *A. tapete* exhibited a marked reduction in their reliance on organic N, a shift attributable to improvements in soil N availability. In this study, the total inorganic N content (NO_3_^−^ + NH_4_^+^) under *A. tapete* was observed to be significantly higher than that in CK soils. Specifically, ammonium N and nitrate N contents were increased by 47.33% and 7.89%, respectively (Table 1), accompanied by significantly higher levels of total N and organic matter. These improvements in nutrient availability contribute to changes in N uptake patterns observed in associated species, with plants in the cushion area showing significantly lower uptake proportions of glycine, and more reliance on inorganic forms, especially NH_4_^+^.

Furthermore, findings from our previous research [35] demonstrate that these differences are varied throughout the growing season. At 4500 m, soil nitrate N content under *A. tapete* was significantly higher than that in CK during both the early and mid-growing seasons, with peak differences observed in mid-season (24.67 μg g^−1^ vs. 15.81 μg g^−1^). Ammonium N content also increased under the cushion, particularly in the early growing season. Simultaneously, *A. tapete* significantly enhanced soil net N mineralization rates and the activity of key N-cycle enzymes, such as nitrate reductase and nitrite reductase, further promoting inorganic N accumulation in the rhizosphere.

Together, these results suggest that the cushion plant *A. tapete* alters both the spatial and temporal patterns of N availability in alpine soils, providing more accessible inorganic N for associated species [17,18,32,35]. As a result, plants growing inside cushion patches experience reduced stress to exploit organic N sources, leading to nutrient niche shifts compared to their conspecifics in CK. This mechanism supports the observed divergence in N uptake strategies and highlights the ecosystem engineering role of *A. tapete* in regulating nutrient dynamics at both community and functional levels.

### 3.3. N Uptake Pattern Changed

Among the three associated species—*C. atrofusca*, *C. incanus*, and *P. saundersiana*—clear shifts in N uptake were observed between CA and CK soils, reflecting their adaptive strategies and ecological niche differentiation. In CK, *C. atrofusca* increased its glycine uptake from 8.93% in CA to 18.69%, while its NH_4_^+^ uptake declined from 44.18% to 25.72%. This adjustment likely reflects the reduced availability of ammonium in CK soils, prompting greater reliance on glycine. Notably, *C. atrofusca* maintained the highest NO_3_^−^ uptake ratio among the three species under both conditions, suggesting a consistent preference for nitrate. Similar nitrate affinity has been reported in other alpine sedges such as *Carex parvula* and *C. alatauensis* [45,46].

In contrast, *C. incanus* exhibited a pronounced shift toward glycine and NO_3_^−^ uptake under CK conditions, with proportions rising from 3.60% to 16.15% and from 24.09% to 47.64%, respectively, while NH_4_^+^ uptake dropped sharply from 72.31% to 36.20%. This pattern highlights the species’ plasticity in nutrient foraging in response to reduced inorganic N availability. Similarly, *P. saundersiana* showed a significant increase in glycine uptake (from 9.56% to 16.71%), whereas changes in NO_3_^−^ (38.91% to 39.29%) and NH_4_^+^ (51.53% to 44.00%) were relatively minor, indicating a more conservative and balanced acquisition strategy [47,48].

Overall, although organic N was relatively abundant, inorganic N remained the primary source for all species. In CA, most species—including *A. tapete*, *C. incanus*, and *P. saundersiana*—showed a clear preference for NH_4_^+^, consistent with its lower energetic cost of assimilation compared with NO_3_^−^ [49,50]. In CK, however, *C. incanus* and *C. atrofusca* shifted toward greater NO_3_^−^ and glycine use, demonstrating the capacity of alpine plants to flexibly adjust nutrient uptake strategies [51,52,53]. Such differentiation in chemical N niches supports coexistence and spatial complementarity (Figure 1). Moreover, the enhanced NH_4_^+^ availability generated by *A. tapete* cushions provides a buffering effect against nutrient limitation, reinforcing facilitative interactions and contributing to the resilience and stability of alpine plant communities [17,32,35].

Our findings reflect conditions at 4500 m, the lower distributional limit of *A. tapete*, and therefore, caution should be exercised when extrapolating these results to other elevations with different environmental conditions. Although we controlled the cushion diameter to reduce morphological variability, we acknowledge that cushion age and density may still influence soil processes and plant nutrient uptake. The 30 cm distance between the CA and CK plots followed standard field practices to minimize root overlap, but potential residual belowground influence cannot be fully excluded. Furthermore, our sampling in late July was designed to capture peak plant activity, but seasonal shifts in nitrogen availability and uptake strategies remain a plausible source of variation. Lastly, while the 4 h post-injection interval is widely used in similar studies to measure immediate nitrogen uptake, this time frame may have conservatively estimated organic nitrogen acquisition, which should be further investigated in future work.

## 4. Materials and Methods

### 4.1. Study Site

The study site is located on the southern slope of the Nyenchenthanglha Mountains in Damxung County (91°05′ E, 30°51′ N), Xizang Autonomous Region, with an experimental station affiliated with the Lhasa Plateau Ecological Research Station, the Chinese Academy of Sciences. This region is characterized by a plateau sub-cold zone monsoon semi-arid climate, with pronounced seasonal differences between dry and wet periods. The region also shows interannual variability, with mean annual precipitation fluctuating between 420 and 520 mm and mean annual temperature varying by approximately 1 °C across different years (data from the Lhasa Plateau Ecosystem Research Station, 2013–2022). The mean annual temperature is 1.3 °C, with the coldest month (January) averaging −10.4 °C and the warmest month (July) reaching 10.7 °C. Average annual precipitation is 476.8 mm, of which 85.1% occurs between June and August. The annual evaporation rate is 1725.7 mm. The soil in this region is classified as alpine meadow soil, with a thickness of approximately 0.3–0.5 m. The granulometric composition of the soil consists of 45% sand, 40% silt, and 15% clay, which may influence nitrogen availability and uptake dynamics. The soil is rich in organic matter and has a pH ranging from 6.2 to 7.7 [4,24].

Alpine meadows dominated by *K. pygmaea* are widely distributed along the mountain slopes, with *A. tapete* interspersed throughout the alpine meadow community. The species is primarily found at elevations ranging from 4500 m to 5200 m. This study selected 4500 m—representing the lower boundary of *A. tapete* distribution—as the experimental site [18]. Based on our field survey, the alpine grassland vegetation at this site included an *A. tapete* coverage of approximately 6.88%. Other dominant species included *Gentiana squarrosa* (~6.40% coverage), *Artemisia desertorum* (~4.58%), and *C. atrofusca* (~4.06%). Additional coexisting species included *C. incanus*, *P. saundersiana*, *Stipa aliena*, and *Anaphalis sp*., contributing to an overall vegetation coverage of about 40% (Figure 2) [18,32,54]. At the 4500 m elevation, the three associated plant species—*C. atrofusca*, *C. incanus*, and *P. saundersiana*—were particularly abundant within and around the cushion plant *A. tapete*. Therefore, these species were selected as the focal plants for this study.

### 4.2. Experimental Layout

In this experiment, the ^15^N isotope labeling method was used to determine the N uptake patterns of associated plants in CA and CK plots during the peak growing season (late July). *C. atrofusca*, *C. incanus*, and *P. saundersiana*—three species commonly found growing within *A. tapete* cushion bodies—were selected as the target plants. A total of 20 pairs of circular plots were randomly selected, with each pair consisting of one plot inside and one plot outside an *A. tapete* individual. The distance between each pair was approximately 30 cm (Figure 3). These circular plots were assigned to four treatment groups: ^15^NH_4_^+^ (injected with ^15^NH_4_NO_3_), ^15^NO_3_^−^ (injected with NH_4_^15^NO_3_), ^15^N-glycine, and a control (CK; injected with distilled water), with five replicates per group. Two criteria were applied when selecting *A. tapete* individuals for sampling: (1) the cushion diameter had to be relatively uniform—approximately 10–20 cm—to minimize potential area effects; (2) the cushion must contain all three associated species (*C. atrofusca*, *C. incanus*, and *P. saundersiana*), each with at least three individuals. Within each plot, individuals of each species were selected to be of similar size to reduce biomass-induced variability in uptake measurements. To ensure comparability, selected individuals differed by <20% in height and canopy diameter and showed no visible stress symptoms. The 30 cm distance between paired CA and CK plots followed previous field practice in this system and was chosen to minimize belowground overlap while maintaining comparable micro-topographic context.

### 4.3. Experimental Process

The experiment was conducted on a sunny day. The ^15^N (^15^NH_4_^+^, ^15^NO_3_^−^, and ^15^N-glycine) was injected into the soil within a 2 cm radius around the target plants at both CA and CK sites. The injection concentration was 0.3 μg N d.w.soil g^−1^ dry soil, meaning that each gram of dry soil received 0.3 μg of ^15^N-labeled N. The control (CK) plots received an equal volume of distilled water. Injections were administered using a 5 mL syringe, delivering 3 mL per injection. The injection depth was 7.5 cm, corresponding to the midpoint of the 0–15 cm soil sampling layer. Due to the rapid turnover of amino acids in soil, plant and soil samples were collected 4 h after ^15^N injection, following established protocols [55,56]. The 4 h window follows established short-term uptake protocols to capture immediate acquisition signals. The ^15^N isotope enrichment for each nitrogen source (NH_4_^+^, NO_3_^−^, and glycine) was set at 10%. This enrichment level is commonly used in nitrogen isotope studies to ensure sufficient labeling of nitrogen sources for accurate measurement of plant uptake.

During sampling, all soil samples and above-ground plant materials were collected. In the laboratory, the target plant species from each sample plot were separated into above-ground parts and roots. The plant samples were first rinsed with clean water, then soaked in 0.5 mmol L^−1^ CaCl_2_ solution for 30 min, and finally washed with deionized water to remove any ^15^N tracer adhering to the plant surfaces. Both above-ground and root tissues were then oven-dried at 75 °C for 48 h, weighed to determine total dry mass, and ground into fine powder using a ball mill (MM2, Fa. Retsch, Haan, Germany). Subsequently, the N content (N%) and atom percent excess of ^15^N (APE) were measured using an isotope ratio mass spectrometer (MAT253, Finnigan MAT, Bremen, Germany).

### 4.4. Soil Sampling

After collecting the plant samples, coarse roots and large gravel pieces were manually removed from the soil samples. The fine roots were subsequently removed using a 60-mesh sieve, and the soil was thoroughly homogenized. The processed soil was then divided into four sub-samples:

Two sub-samples were used for determining soil moisture content and pH.

One sub-sample was shaken with 2 M KCl solution for 30 min, followed by filtration through qualitative filter paper to obtain soil extracts [57]. These extracts were used to measure nitrate N (NO_3_^−^-N), ammonium N (NH_4_^+^-N), and glycine concentrations.

The final sub-sample was air-dried and finely ground using a 100-mesh sieve for analysis of total soil carbon and N contents.

NO_3_^−^ and NH_4_^+^ concentrations in the soil extracts were determined using an AA3 continuous flow analyzer (Seal Analytical, Norderstedt, Germany), while amino acid (glycine) content was quantified using an amino acid analyzer (Agilent Technologies, Waldbronn, Germany).

### 4.5. N Uptake Calculation

^15^N uptake (mg ^15^N m^−2^) of associated plant species (U_labelled_) was calculated by multiplying N content (mg N g^−1^ d.w.), biomass (g m^−2^), and APE. APE was calculated as the atom% ^15^N difference between plants from ^15^N treated and CK. The calculation formula is as follows:APE = ^15^Natom%_r_ − ^15^Natom%_c_U_labelled_ = N% × Biomass × APE

The uptake of the three forms of N (^15^NH_4_^+^, ^15^NO_3_^−^, ^15^N-glycine) by *A. tapete* and its associated plants (Unlabelled) was calculated according to the method of McKane et al. [36]:U_unlabelled_ = U_labelled_(m_unlabelled_/m_labelled_) where m_labelled_ is the total mass (g m^−2^) of ^15^N-labeled N injected per plot, m_unlabelled_ is the mass of available N species measured in soil. U_labelled_ is the uptake (g m^−2^) of ^15^N from the source m_labelled_, and U_unlabelled_ is the uptake of available N from the source m_unlabelled_.

Data analysis, statistical testing, and chart plotting were conducted by R studio software (version 4.5.1) and Excel. One-way ANOVA was employed to compare differences in basic soil physicochemical properties between CA and CK, as well as differences in the uptake of the three different N forms by associated plants. Prior to statistical analysis, normality and homogeneity of variance tests were performed to ensure data suitability for parametric testing. In all analyses, a threshold of *p* < 0.05 was considered statistically significant. One-way ANOVA was selected because paired plots were independent sampling units and the data met normality and homoscedasticity assumptions. We further note that the direction and significance of the main effects are robust under reasonable alternative modeling choices.

## 5. Conclusion

*Androsace tapete*, a keystone cushion plant species in the alpine ecosystems of the QTP, plays a pivotal role as an ecological engineer by substantially regulating the N uptake strategies of its associated plants. Through improving soil properties—such as increasing total carbon, total N, and ammonium concentrations—*A. tapete* effectively mitigates the limitation of inorganic N availability in alpine environments. Moreover, it reduces the reliance of associated plants on organic N sources and promotes the uptake of inorganic forms, particularly NH_4_^+^. This shift in N uptake strategy not only represents a physiological adaptation of plants to nutrient-poor conditions but also reflects the broader ecological function of cushion plants in facilitating community coexistence. Collectively, these results indicate that cushion-induced improvements in soil conditions reconfigure plant chemical niches by shifting the relative use of NH_4_^+^, NO_3_^−^, and organic N, thereby reducing direct competition and stabilizing coexistence. Framed as ecosystem engineering, this mechanism links microhabitat modification to community-level diversity and resilience, offering practical insight for alpine conservation and restoration.

## Figures and Tables

**Figure 1 plants-14-03232-f001:**
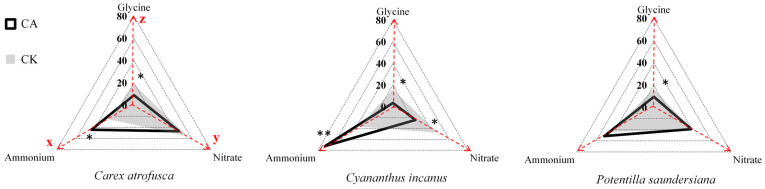
Chemical N niches of associated species within cushion patches (CA) and in adjacent grassland (CK). Axes denote the proportional contributions (%, summing to 100%) of NH_4_^+^ (*x*-axis), NO_3_^−^ (*y*-axis), and glycine (*z*-axis) to total N uptake. Polygons represent the mean niche centroid ± 1 SE for each species and habitat (solid black: CA; gray shaded: CK). Overlapping areas indicate shared chemical niches, whereas polygon shifts reflect niche differentiation between CA and CK. Asterisks denote significant CA–CK differences for a given N form (*p* < 0.05 for *, *p* < 0.01 for **); otherwise, differences should be interpreted as non-significant trends.

**Figure 2 plants-14-03232-f002:**
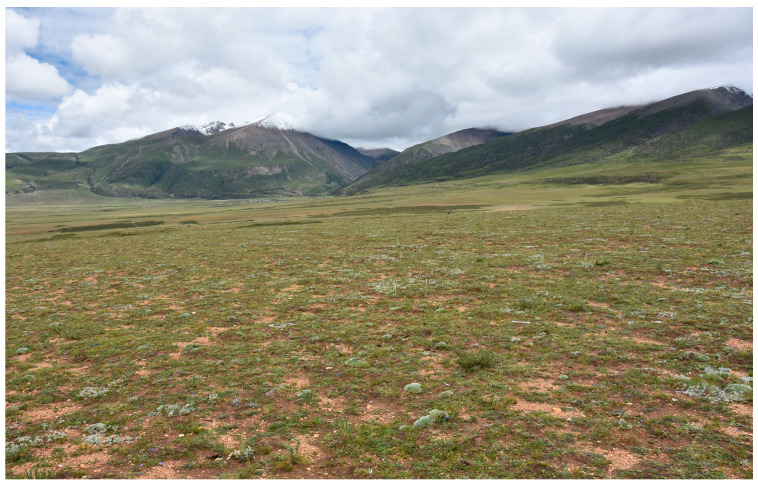
The experiment sites are at Damxung (4500 m).

**Figure 3 plants-14-03232-f003:**
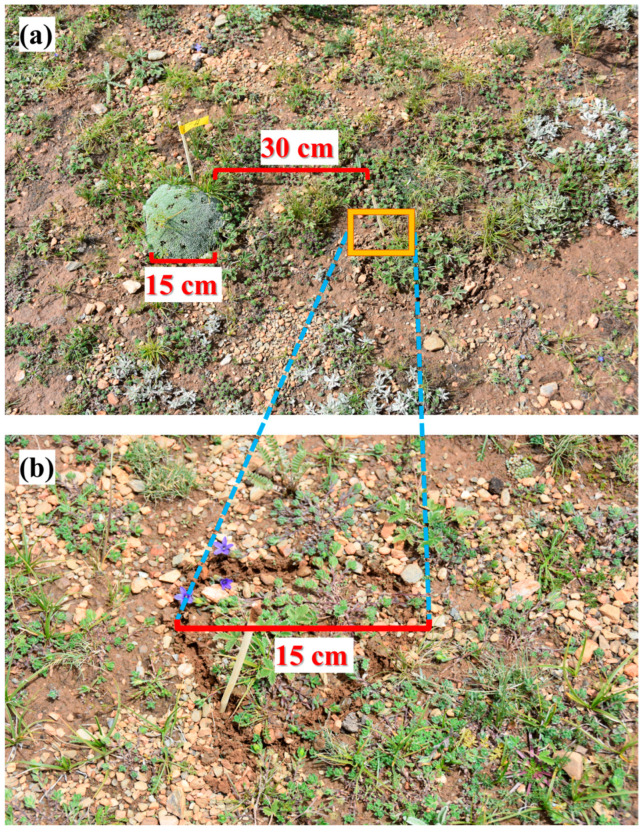
Experimental sites (inside and outside of *A. tapete*). (**a**) shows that the overall photo of experimental plots, with an *A. tapete* (diameter of 15 cm). The CK is 30 cm away from *A. tapete*. (**b**) shows a magnified photo of CK, with a diameter approximately equal to that of *A. tapete* sites.

**Table 1 plants-14-03232-t001:** Soil analysis between CA and CK (0~15 cm). Means ± SE are presented (n = 5).

Soil Properties	CA	CK	Difference (%)
Bulk density (g cm^−3^)	0.62 ± 0.02 b	1.05 ± 0.10 a	−40.95
pH (H_2_O)	6.33 ± 0.04 a	7.03 ± 0.22 a	−9.96
Water content (%)	24.23 ± 1.38 a	22.94 ± 1.25 a	5.62
Total C (g C kg^−1^ d.w. soil)	35.05 ± 2.04 a	23.71 ± 4.12 b	47.82
Total N (g N kg^−1^ d.w. soil)	2.65 ± 0.39 a	1.88 ± 0.07 b	40.96
C/N	13.23 ± 0.26 a	12.61 ± 0.19 a	4.92
Glycine (μg N g^−1^ d.w. soil)	0.016 ± 0.002 a	0.014 ± 0.001 a	14.29
nitrate N (μg N g^−1^ d.w. soil)	0.369 ± 0.16 a	0.342 ± 0.12 a	7.89
ammonium N (μg N g^−1^ d.w. soil)	1.547 ± 0.13 a	1.050 ± 0.25 b	47.33

Different letters indicate significant differences between CA and CK at a 0.05 error probability level.

**Table 2 plants-14-03232-t002:** The biomass between CA and CK. Means ± SE are presented (n = 5).

Plant Species	Biomass (g m^−2^)			
	Leaf		Root	
	CA	CK	CA	CK
*C. atrofusca*	0.43 ± 0.31 a	0.18 ± 0.02 b	1.34 ± 0.34 b	2.96 ± 0.76 a
*C. incanus*	0.49 ± 0.18 a	0.43 ± 0.16 a	0.11 ± 0.01 a	0.16 ± 0.02 a
*P. saundersiana*	0.30 ± 0.20 a	0.25 ± 0.04 a	0.12 ± 0.01 a	0.23 ± 0.12 a

Different letters indicate significant differences between CA and CK at a 0.05 error probability level.

**Table 3 plants-14-03232-t003:** Percent of N uptake by associated plants from three N forms in CA and CK. Means ± SE are presented (n = 5).

Plant Species	Sampling Plots					
	CA			CK		
	Glycine	NO_3_^−^	NH_4_^+^	Glycine	NO_3_^−^	NH_4_^+^
*A. tapete*	7.38 ± 1.25 b	9.78 ± 1.54 b	82.84 ± 4.62 a	-	-	-
*C. atrofusca*	8.93 ± 1.25 a	46.89 ± 10.23 a	44.18 ± 9.82 a	18.69 ± 4.35 b	55.59 ± 11.06 a	25.72 ± 5.63 b
*C. incanus*	3.60 ± 0.63 a	24.09 ± 6.45 a	72.31 ± 13.69 a	16.15 ± 3.56 b	47.64 ± 8.69 b	36.20 ± 6.12 b
*P. saundersiana*	9.56 ± 1.32 a	38.91 ± 7.69 a	51.53 ± 11.47 a	16.71 ± 4.12 b	39.29 ± 7.52 a	44.00 ± 10.65 a

Different letters indicate significant differences between CA and CK at a 0.05 error probability level.

## Data Availability

All data generated or analyzed during this study are included in the published article and also available from the corresponding author on reasonable request.

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
