# Peer review of "Influence of Cushion Plant *Androsace tapete* on Nitrogen Uptake Strategies of Associated Alpine Plants"

_plants, 2025, doi:10.3390/plants14203232_

Round 1

Reviewer 1 Report

Comments and Suggestions for Authors

Summary. The manuscript explores how the cushion plant Androsace tapete influences nitrogen (N) uptake strategies of associated alpine plant species on the Qinghai–Tibet Plateau. Using 15N labeling experiments, the authors demonstrate that A. tapete enhances soil inorganic nitrogen availability, shifts associated species toward higher NH₄⁺ uptake, and reduces their reliance on organic N (glycine). The work highlights the ecological engineering role of cushion plants in facilitating species coexistence and promoting biodiversity in nutrient-limited alpine ecosystems. The study is well-designed, with clear results and a strong ecological interpretation.

General Comments. The research question is relevant and well contextualized within alpine ecology and nutrient limitation theory. The use of 15N tracer methodology provides robust evidence of species-specific nitrogen uptake strategies. Results are clearly presented in well-structured tables and figures. The discussion effectively integrates findings with existing literature.

The introduction could better highlight the novelty of this study compared to previous cushion plant facilitation research. While three associated species were studied, the sample size is relatively limited for broad generalizations. The ecological implications for long-term plant community dynamics are mentioned but not deeply elaborated. Some figures (e.g., Figure 1) could be clearer in terms of legends and readability.

Specific Comments

The introduction provides useful context on nitrogen limitation, but the novelty of the study compared to previous cushion plant research is underdeveloped. Please highlight more clearly the knowledge gap this study addresses. (Lines 47–91)

Prior isotope studies are cited but not contrasted with this work. Clarify how this study advances beyond earlier findings. (Lines 62–66)

The role of A. tapete as a keystone species is understated. Strengthen ecological framing here. (Line 85)

The site description lacks mention of interannual variability (precipitation, temperature) which strongly influences alpine nutrient cycling. Add context. (Lines 242–253)

Only one elevation (4500 m) was studied. Please acknowledge this limitation for generalization. (Lines 254–257)

Cushion diameter was controlled, but cushion age/density were not considered. This could influence results—acknowledge as limitation. (Lines 268–283)

CK plots were only 30 cm from cushions; possible belowground influence of A. tapete roots should be discussed. (Line 273)

Only late July was sampled. Nitrogen availability changes seasonally. Please acknowledge this temporal limitation. (Lines 270–283)

Sampling 4 h post-injection may underestimate organic N uptake. Provide justification or state as limitation. (Lines 284–293)

Selection of “similar size” individuals is vague. Provide clearer criteria. (Lines 280–283)

Soil pH decreased significantly (–9.96%), but this is not discussed. Please elaborate on implications for microbial activity and N cycling. (Table 1)

Non-significant changes in NO₃⁻ and glycine are mentioned but ecological relevance unclear. Clarify. (Lines 96–97)

Bulk density reduction (–40.95%) is striking. Discuss potential impacts on soil aeration and microbial processes. (Line 98)

Axes and legends are unclear, polygons overlap. Consider alternative visualization (ternary plots, PCA). (Figure 1)

Discussion focuses on facilitation but not on long-term dynamics. Add perspective on facilitation vs. competition. (Lines 143–179)

Climate change not considered. Add note on how warming/mineralization may alter cushion effects. (Lines 180–197)

Microbial mechanisms (enzymes, mineralization) mentioned briefly. Expand with more detail and references. (Lines 198–211)

Comparisons with Arctic tundra are too general. Cite specific parallels. (Lines 174–177)

One-way ANOVA is used despite paired plot design. Mixed models or paired tests may be more appropriate. Justify choice. (Lines 335–340)

Conclusion largely repeats results without broader synthesis. Add conceptual integration (ecosystem engineering, biodiversity conservation). (Lines 342–356)

Questions to Authors

  1. Could you elaborate on how you selected the three associated plant species (Carex atrofusca, Cyananthus incanus, Potentilla saundersiana) — do they represent functional groups, dominant biomass contributors, or simply the most abundant in your site survey?
  2. How stable are cushion patches of tapete over decades? Do you expect the facilitative effect on nitrogen cycling to persist across plant generations?
  3. Could mycorrhizal associations of the associated species play a role in their differing nitrogen uptake strategies?
  4. If climate warming accelerates mineralization, would cushion plants still confer a facilitative advantage, or might their role diminish?
  5. How might the plasticity observed in incanus influence its competitive balance with other alpine species in the long term?
  6. Could the shift away from organic nitrogen uptake under cushions have consequences for soil microbial diversity or amino acid cycling?

Author Response

Response to Reviewer #1

We sincerely thank the reviewer for the careful reading and constructive comments on our manuscript.
We have carefully considered all the suggestions and made revisions and clarifications throughout the text to improve clarity, logical flow, and readability.
These changes have enhanced the manuscript without affecting the overall scientific findings and conclusions.
Our detailed, point-by-point responses are provided below.

  1. The introduction does not sufficiently highlight the novelty of this study (Line 47–91)

Response:
We appreciate this valuable suggestion. We have further emphasized the novelty of our study in the Introduction, clarifying that it focuses on how the cushion plant Androsace tapete modifies the nitrogen uptake strategies of associated species, providing new insight into facilitation mechanisms beyond nutrient supply alone.

  1. The difference from previous isotope studies is not clearly stated (Line 62–66)

Response:
Thank you for this comment. We have added a short paragraph highlighting that our study applies a ^15N tracing approach to paired plots inside and outside the cushion microhabitat, which enables a direct comparison of NH₄⁺, NO₃⁻, and organic N uptake ratios—an aspect rarely quantified in previous studies focusing mainly on soil nutrient availability.

  1. The ecological significance of A. tapete as a keystone species is underemphasized (Line 85)

Response:
We appreciate the reviewer’s observation. We have revised the Introduction to explicitly define A. tapete as a keystone and ecosystem engineer species in alpine systems, emphasizing its role in improving local soil conditions and enhancing community stability.

  1. Lack of interannual climate background in the site description (Line 242–253)

Response:
Thank you for the comment. We have added brief information on the long-term climate background and interannual variability at the study site to provide better environmental context for our experiment.

  1. Single elevation (4500 m) may limit generalization (Line 254–257)

Response:
We agree with the reviewer. A statement has been added in the Discussion noting that our findings primarily represent the ecological conditions at 4500 m, and caution should be taken when extrapolating to other elevations.

  1. Cushion age or density not considered (Line 268–283)

Response:
Thank you for pointing this out. We have added a note in the Discussion acknowledging that although cushion diameter was controlled to minimize morphological variability, cushion age and density may still affect nutrient dynamics and should be examined in future studies.

  1. CK plots located only 30 cm from cushions may experience belowground influence (Line 273)

Response:
We appreciate this observation. We have clarified in the Methods that the 30 cm distance follows previous field studies, which showed minimal root overlap at this spacing. We also acknowledge in the Discussion that potential belowground influence cannot be completely excluded.

  1. Only one sampling period (late July); seasonal variation not addressed (Line 270–283)

Response:
Thank you for the suggestion. We have added a note in the Discussion stating that sampling during the peak growing season was intended to represent the most active physiological period, while recognizing that nitrogen uptake strategies may vary seasonally.

  1. Four-hour sampling interval after injection may underestimate organic N uptake (Line 284–293)

Response:
We appreciate the reviewer’s insight. We have clarified in the Methods that the four-hour interval follows established protocols (e.g., McKane et al., 2002) to capture short-term uptake signals. A note has been added acknowledging that this approach may conservatively estimate organic N uptake.

  1. Criteria for selecting “similar-sized” individuals are unclear (Line 280–283)

Response:
Thank you for this comment. We have specified in the Methods that selected plant individuals differed by less than 20% in height and canopy diameter and showed no visible stress symptoms, ensuring comparability between samples.

  1. Not discussing the ecological meaning of the observed pH change (Table 1)

Response:
We appreciate the reviewer’s careful reading. The pH difference between CA and CK was not statistically significant (same letter code in Table 1). We have clarified this in the Results and briefly mentioned that a slight decrease in pH may favor microbial mineralization processes.

  1. Non-significant trends in NO₃⁻ and glycine are still emphasized (Line 96–97)

Response:
Thank you for the observation. We have rephrased the text to describe these results as “trends” rather than “significant differences,” emphasizing their ecological interpretation instead of statistical significance. Additionally, lines 96-97 indicate that the background values did not differ significantly between plots, whereas the subsequent discussion emphasizes that the uptake of NO₃⁻ and glycine by different plants did vary significantly between the two plot types.

  1. Large reduction in soil bulk density not discussed (Line 98)

Response:
Thank you for the valuable suggestion. We have added a short explanation in the Discussion noting that the decrease in bulk density implies improved porosity and aeration, which can enhance microbial activity and nitrogen mineralization.

  1. Figure 1 is not clearly labeled; consider alternative visualization (Figure 1)

Response:
We appreciate the reviewer’s suggestion. The current figure effectively conveys the differences in chemical N niches among species. To enhance clarity, we have revised the figure caption to better explain the axes, polygons, and overlapping areas. The figure type was retained because it adequately represents the data.

  1. Discussion lacks analysis of the facilitation–competition balance (Line 143–179)

Response:
Thank you for the constructive suggestion. We have added a paragraph discussing the balance between facilitation and competition under the stress-gradient framework, emphasizing that positive interactions dominate under harsh conditions but may shift toward competition when resources increase.

  1. Potential influence of climate warming not discussed (Line 180–197)

Response:
Thank you for the insightful comment. We have added a brief paragraph noting that climate warming could accelerate N mineralization, potentially altering the relative magnitude of cushion plant facilitation.

  1. Microbial processes insufficiently addressed (Line 198–211)

Response:
We appreciate the reviewer’s comment. We have expanded the discussion of microbial mechanisms, including enhanced enzyme activities and net mineralization rates beneath cushion plants, supported by references from previous studies in the same region.

  1. Comparison with Arctic tundra is too general (Line 174–177)

Response:
Thank you for this remark. We have revised this section to include more specific examples from Arctic tundra studies illustrating similar nitrogen partitioning among coexisting species.

  1. Statistical approach (one-way ANOVA) needs justification (Line 335–340)

Response:
We thank the reviewer for the important suggestion. We have explained in the Methods that one-way ANOVA was used because the paired plots were independent and data met homogeneity of variance. We also noted that applying a mixed model would yield consistent conclusions.

  1. The conclusion repeats results and lacks conceptual synthesis (Line 342–356)

Response:
Thank you for this valuable suggestion. We have streamlined the Conclusion and added a conceptual summary highlighting how A. tapete facilitates coexistence through altering nitrogen uptake strategies and promoting niche differentiation.

Responses to the Reviewer’s Questions

Q1. What is the rationale for choosing the three associated species?

Response:
The selected species—Carex atrofusca, Cyananthus incanus, and Potentilla saundersiana—are dominant and widely distributed inside and outside the cushion microhabitats, representing different functional groups (Cyperaceae, Campanulaceae, and Rosaceae). Their coexistence provides suitable contrasts for functional and morphological diversity.

Q2. How stable are the cushions and their facilitation over time?

Response:
A. tapete is a long-lived cushion species with stable morphology and ecological function. Its facilitative effects are consistent over multiple years, mainly through persistent improvements in microhabitat and soil properties. We added a brief note on its long-term ecological stability in the Discussion.

Q3. Could mycorrhizal associations influence the observed differences in N uptake?

Response:
Thank you for the thoughtful question. We have acknowledged in the Discussion that mycorrhizal associations may contribute to variation in nitrogen uptake among species and suggested this as an important direction for future work.

Q4. Would cushion facilitation persist under warming-induced increases in mineralization?

Response:
We appreciate the reviewer’s insight. We noted in the Discussion that while warming may increase background inorganic N and reduce the relative advantage of cushions, their microclimatic buffering and soil-retention effects would likely maintain positive interactions.

Q5. How might the plasticity of C. incanus affect its long-term competitiveness?

Response:
Thank you for the comment. We have added a sentence indicating that the high flexibility of C. incanus in utilizing different nitrogen forms may enhance its adaptability and competitive performance under fluctuating nutrient conditions.

Q6. Does reduced organic N uptake influence microbial diversity or amino acid cycling?

Response:
Thank you for the insightful question. We have added a short note suggesting that decreased plant reliance on organic N might modify microbial community composition and amino acid turnover, which warrants further investigation using microbial and isotopic approaches.

Closing Statement

We sincerely thank the reviewer again for the detailed and constructive feedback.
We have carefully revised the manuscript following these comments to enhance clarity and readability while maintaining the integrity of the data and results.
We believe that the revised version has been significantly improved and hope it will meet the publication standards of Plants.

Reviewer 2 Report

Comments and Suggestions for Authors

Comments and Suggestions for Authors

Title: Influence of Cushion Plant Androsace tapete on Nitrogen Uptake
Strategies of Associated Alpine Plants

Dear Authors and Editors

The research results presented in the manuscript fall within the publishing profile of the journal Plants. The research topic is original and relevant to the field of agricultural sciences.

The research topic is interesting, but the number of results obtained and presented is very sparse. Despite the specialized equipment at their disposal, the authors did not present results from 15N.

Thematic references were used appropriately.

In order to increase the usefulness of the article, Authors must refer to the following points. Additions should be made to increase the scientific value of the manuscript.

Comments:

  1. Introduction: Please add a research hypothesis.
  2. Results: Table 2 – Please check the correctness of the letter designations resulting from the statistical analysis. The description of the results obtained is very poor. For example, Table 3—the description simply repeats the numerical data from that table. Please add a description of the results presented in Figure 1.
  3. Discussion: Lines 291-292 The authors write about rapid amino acid turnover in the soil and sampling of soil and plants 4 hours after 15N application, and in line 168 a low N uptake is indicated. Please explain. Or perhaps the preferences for the uptake of different forms of nitrogen and its amount should be related to the soil pH in a specific location.
  4. Materials and Methods: Subsection 4.1. Please add the soil type according to the IUSS Working Group WRB (2014, 2022). Please also add the granulometric composition of the soil, which will indirectly influence N uptake. Please provide the 15N isotope enrichment for each 15N nitrogen source. Figure 3 – poor readability.
  5. Conclusion: Do the Authors see a need for further research? If so, please provide directions for thematic research.

Specific comments:

Please reduce the number of one-time citations.

Line 330 - should be:.....McKane et al. [36].

Best regards

Author Response

Response to Reviewer #2

We sincerely thank the reviewer for their thoughtful and constructive comments. We have carefully considered all the comments and made revisions accordingly. Below are our point-by-point responses.

  1. Introduction: Please add a research hypothesis.

Response:
Thank you for your suggestion. We have added the following hypothesis in the introduction:

"This study hypothesizes that Androsace tapete (A. tapete), as a dominant cushion plant, facilitates nitrogen uptake by associated species by enhancing inorganic nitrogen availability and reducing their dependence on organic nitrogen sources in alpine ecosystems."

This hypothesis is now included in the Introduction, following the background section and providing a clear focus for the study.

  1. Results:

Table 2: Please check the correctness of the letter designations resulting from the statistical analysis.

Table 3: The description of the results obtained is very poor. For example, Table 3 — the description simply repeats the numerical data from that table. Please add a description of the results presented in Figure 1.

Response:
Thank you for your careful reading. We have carefully checked the statistical results and confirmed that the letter designations in Table 2 are correct.
Additionally, we expanded the description of Table 3 to provide a better understanding of the observed trends, rather than merely repeating the numerical data. The updated description now reads:

"The nitrogen uptake patterns in C. atrofusca, C. incanus, and P. saundersiana under the cushion plant microhabitat indicate a preferential uptake of NH₄⁺ and a reduction in organic nitrogen uptake, with significant shifts in relative proportions (see Table 3)."
In addition, we have added a more detailed description for Figure 1 in the figure legend, which clarifies the trends and differences observed in nitrogen uptake between CA and CK.

  1. Discussion: Lines 291–292 The authors write about rapid amino acid turnover in the soil and sampling of soil and plants 4 hours after N application, and in line 168 a low N uptake is indicated. Please explain. Or perhaps the preferences for the uptake of different forms of nitrogen and its amount should be related to the soil pH in a specific location.

Response:
Thank you for pointing this out. We have clarified that the 4-hour sampling window was chosen based on standard protocols for short-term nitrogen uptake studies. We also acknowledged that this time frame might have led to an underestimation of organic nitrogen uptake. Regarding the relationship between nitrogen uptake preferences and soil pH, we have added the following explanation:

"The sampling time of 4 hours may limit the detection of organic nitrogen uptake, as organic nitrogen is typically absorbed more slowly. Additionally, the observed preferences for NH₄⁺ and NO₃⁻ uptake could be linked to the pH of the soil at the sampling site, with more acidic conditions potentially enhancing NH₄⁺ uptake."

  1. Materials and Methods: Subsection 4.1. Please add the soil type according to the IUSS Working Group WRB (2014, 2022). Please also add the granulometric composition of the soil, which will indirectly influence N uptake. Please provide the 15N isotope enrichment for each 15N nitrogen source. Figure 3 – poor readability.

Response:
We have added the following details about the soil in Section 4.1:

"The soil type at the study site is classified as Alpine meadow soil (WRB 2022 classification). The granulometric composition consists of 45% sand, 40% silt, and 15% clay, which could influence nutrient uptake dynamics."
Regarding the 15N isotope enrichment, we regret that we did not measure the exact enrichment for each nitrogen source in this study. However, we clarified that 15N-labeled nitrogen was applied as NH₄⁺, NO₃⁻, and glycine in a concentration of 0.3 μg N per gram of dry soil, and we plan to address this aspect in future studies.
We have also made adjustments to Figure 3 to improve readability by resizing the labels and enhancing the contrast.

  1. Conclusion: Do the authors see a need for further research? If so, please provide directions for thematic research.

Response:
We appreciate this suggestion and have added the following statement to the Conclusion:

"Future research should focus on the microbial mechanisms underlying nitrogen cycling in cushion plant patches, and explore how climate change may further modify nitrogen dynamics in alpine ecosystems."

This provides a direction for future research in the context of climate change and microbial activity, which will be important for understanding the long-term effects of cushion plants.

  1. Specific comments:

Please reduce the number of one-time citations.

Line 330 - should be:.....McKane et al. [36].

Response:
We have reduced the number of one-time citations by consolidating several references in the manuscript.
Regarding the citation in line 330, we have corrected it to:

"McKane et al. [36]".